# MRI-guided robotic arm drives optogenetic fMRI with concurrent Ca$^{2+}$ recording

Yi Chen[1,2], Patricia Pais-Roldan[1,2], Xuming Chen[1,3], Michael H. Frosz [4] & Xin Yu [1,5]

Optical fiber-mediated optogenetic activation and neuronal Ca$^{2+}$ recording in combination with fMRI provide a multi-modal fMRI platform. Here, we developed an MRI-guided robotic arm (MgRA) as a flexible positioning system with high precision to real-time assist optical fiber brain intervention for multi-modal animal fMRI. Besides the ex vivo precision evaluation, we present the highly reliable brain activity patterns in the projected basal forebrain regions upon MgRA-driven optogenetic stimulation in the lateral hypothalamus. Also, we show the step-wise optical fiber targeting thalamic nuclei and map the region-specific functional connectivity with whole-brain fMRI accompanied by simultaneous calcium recordings to specify its circuit-specificity. The MgRA also guides the real-time microinjection to specific deep brain nuclei, which is demonstrated by an Mn-enhanced MRI method. The MgRA represents a clear advantage over the standard stereotaxic-based fiber implantation and opens a broad avenue to investigate the circuit-specific functional brain mapping with the multi-modal fMRI platform.

[1] Research Group of Translational Neuroimaging and Neural Control, High-Field Magnetic Resonance, Max Planck Institute for Biological Cybernetics, 72076 Tuebingen, Germany. [2] Graduate Training Centre of Neuroscience, University of Tuebingen, 72076 Tuebingen, Germany. [3] Department of Neurology, Renmin Hospital of Wuhan University, Wuhan University, 430060 Wuhan, China. [4] Max Planck Institute for the Science of Light, 91058 Erlangen, Germany. [5] Athinoula A. Martinos Center for Biomedical Imaging, Massachusetts General Hospital and Harvard Medical School, Charlestown, MA 02129, USA. Correspondence and requests for materials should be addressed to X.Y. (email: xin.yu@tuebingen.mpg.de)

A multi-modal brain mapping platform for animals has been established by merging the fiber optic-mediated optogenetic activation and neuronal Ca[2+] recording with functional magnetic resonance imaging (fMRI)[1–5]. Given its non-magnetic properties, the optical fiber can be used in combination with fMRI brain mapping without electromagnetic interference with the radio frequency (RF) transmission and magnetic gradient switching of the MR scanner[2,3,6,7]. The increased cellular specificity of genetic labeling reassures the advantageous usage of optical fiber recording/imaging to track neural spiking activity in the deep brain regions[8–13]. However, one emerged challenge is how to precisely target specific functional nuclei in the animal brain[8,14]. The procedure of fiber optic implantation in rodent studies has been commonly performed with conventional stereotaxic devices[2,3,7–11,14,15], but the success rate to precisely target the deep brain nuclei remains low, especially for the functional nuclei that cover only a few hundred microns space in the animal brain, e.g., the central thalamic nulcei[8]. A solution to precisely target the genetically labeled neuronal tracts or subdivisions of functional nuclei could significantly improve the reproducibility of basic scientific discoveries. Here, we report an MRI-guided robotic arm (MgRA) positioning device to maneuver the real-time fiber optic implantation into the animal brain inside a high-field MR scanner (14.1 T), intended for parallel optogenetics and/or calcium imaging and fMRI studies.

The genetic expression of channelrhodopsins (ChR2) has been extensively applied to target-specific cell types in the deep brain nuclei, such as the dopaminergic neurons in the midbrain[9], the orexin in the lateral hypothalamus (LH)[16,17] or noradrenergic neurons in the locus coeruleus[18]. The cell-type specific genetic labeling ensures the optogenetic activation on neuronal ensembles of interest assuming that the optical fiber is precisely located at the functional nuclei. However, the stereotaxic device-driven fiber optic implantation scheme shows little flexibility after the fiber tips are fixed in the brain for either fMRI mapping, electrophysiological recordings, or behavioral studies[8,19]. The precise coordinates of a certain functional brain nucleus can vary between different animals, and incorrect positioning may result in largely altered functional activation and behavioral outcomes. This systematic error, which is intrinsic to the blind optical fiber placement, can potentially conceal important discoveries and lead to inappropriate conclusions in causality analysis. Using MgRA assisted fiber-optic insertion in combination with real-time fMRI, we can provide a step-wise optogenetic activation scheme to allow multi-site targeting through a fiber insertion trajectory during the fMRI study. This strategy can not only improve the precision, but also provide a thorough view to examine the subtle differences in the whole brain activation patterns when targeting the sub-regions of the functional nuclei of interest.

Numerous efforts have been made to develop robotic positioning systems inside the MRI scanner for translational application from animals to the clinical practice, e.g., deep brain stimulation or brain tumor ablation[20–26]. In contrast to the growing access to robotic manipulation strategies inside large-bore MRI scanners (e.g., 1.5 or 3 T human scanner), there are only a handful of works that have implemented remote controlling systems inside high field MRI scanners with smaller bore (>7 T, <12 cm gradient bore size), which have been applied to adjust sample orientation within the $B_0$ field[27] or to tune RF coil arrays[28]. To the best of our knowledge, there is currently no MRI-compatible robotic control system to assist fiber optic insertion in small bore high field MRI scanners (>9.4 T) for optogenetic fMRI studies. Hence, as a proof-of-concept, we developed an MgRA to provide a flexible positioning system inside a 14.1 T MRI scanner which assists fiber optical brain intervention in animals. Besides an ex vivo precision evaluation, we present a series of in vivo studies showing the whole brain activity patterns upon optogenetic stimulation of MgRA-targeted nuclei in the LH or thalamus in a step-wise manner and with simultaneous fiber-optic calcium recordings to specify the region-specific optogenetic activation patterns. In addition, the MgRA system can be applied for region-specific deep brain microinjection. Here, we demonstrate a series of high precision brain interventional applications in the context of multi-modal neuroimaging using the MgRA system.

## Results

**Mechanical design of the MgRA with ex vivo operation.** A stepper motor-driven MgRA was designed for real-time control of the insertion of an optical fiber into animal brains inside a 14.1

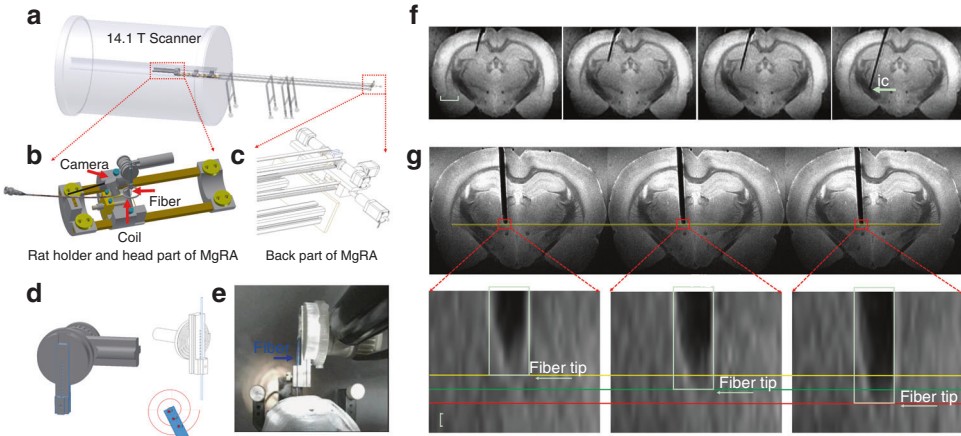

**Fig. 1** 3D view of the MgRA and its application in ex vivo studies. **a** Overview of the MgRA inside the 14.1 T MR scanner. **b** Schematic of the customized animal holder and head part of the MgRA. Both MR compatible camera and surface transceiver coil are included for monitoring the fiber optic insertion inside the MR scanner. **c** Stepper motors implemented at the back part of the MgRA to control up to four degrees-of-freedom movement. The long arm reaching 4.7 m away from the magnetic center point excludes the influence by the ultra-high magnetic field. **d** Schematic drawing of the Archimedean spiral design to transmit the dorsal–ventral movement. **e** Snapshot of the mechanically controlled fiber optic movement videotaped by the built-in camera. **f** Time-lapsed images showing the optical fiber targeting the hippocampus, thalamus, and internal capsule along the insertion trajectory. Scale bar, 2 mm. **g** Three continuous MRI anatomical images with step distance 50 μm (the MRI in-plane resolution is 50 × 50 μm², thus it can be seen that the distance moved in each step is approximately 50 μm). Scale bar, 50 μm

T scanner (Fig. 1a, 3D schematic view in Supplementary Movie 1, Supplementary Fig. 1). The MgRA contains two key parts: the front part (head of the MgRA) includes the driving pieces and a customized rat holder (Fig. 1b), and the back part accommodates the stepper motors to fulfill the optical fiber movement with multi-degree of freedom (Fig. 1c). The coupling of the actuator (back part) to the matching toothed pulley in the head was achieved by a synchronous belt drive (Fig. 1c) in a form-fit manner, without slippage and run at constant speed. Insertion of the optical fiber in the dorsal-ventral direction into the rat brain is executed using an Archimedean spiral mechanism to achieve high precision and accuracy (Fig. 1d). With a built-in MRI compatible camera, the insertion of the optical fiber could be monitored outside of the scanner to verify the effectiveness, safety, and feasibility of the MgRA (Fig. 1e and Supplementary Movie 2), simultaneously tracked with anatomical MRI. The assembly of all components provides the MgRA unique features in a portable frame that can be easily located inside the MRI room substituting the conventional subject table. A more detailed description of MgRA can be found in Methods and Supplementary Figs. 11–13.

The MgRA was first evaluated in perfused brains embedded in agarose (Fig. 1f), in order to simulate the procedure of intracerebral fiber insertion in the living animal. The optical fiber was first inserted into the agarose-embedded brain preparation in a 100 μm step-wise manner, and real-time MRI images were acquired to monitor the movement trajectory and to identify the location of the fiber tip (Supplementary Movie 3). Precision of the MgRA was determined as the smallest step in the dorsal–ventral direction that could be maneuvered based on the remote stepper motor controlling. Figure 1g shows the step-wise movement of the fiber inside the rat brain at 50 μm per step with high-resolution MRI time-lapsed 2D images (Supplementary Movie 4). It is worth noting that fiber insertion trajectories can be optimized with special angles to target specific deep brain nuclei or fiber bundles while avoiding disturbance of neural circuits, projection pathways of interest or certain brain vessels. For instance, an angled fiber optic insertion can be implemented to target the internal capsule to preserve the ascending pathway of the thalamocortical circuits (Fig. 1f and Supplementary Fig. 2). In summary, MgRA-based fiber optic insertion in the ex vivo brain verifies its functionality and demonstrates the stability in terms of remote motor control.

**In vivo MgRA-driven fiber insertion with optogenetic fMRI.** MgRA allows the insertion of optical fibers in vivo inside the 14 T MRI scanner, which induces great advantages for optogenetic fMRI studies[3,29,30]. To locate the fiber tip prior to intracerebral insertion inside the MRI scanner, two procedures were followed. First, we implemented two MRI-compatible cameras to visually locate the fiber tip, as well as the craniotomy on the animal skull (Fig. 2a and Supplementary Movie 5). Second, a prior application of a manganese-treated agarose gel was applied over the skull and the sequential lowering of the fiber was monitored with real-time anatomical MRI to locate the fiber tip as well as the craniotomy hole on the skull to guide the fiber targeting inside the brain (Supplementary Fig. 3). A more detailed description can be found in Methods, Supplementary Fig. 3. Figure 2b shows snapshots of the fiber tip outside the brain during the MgRA-driven fiber insertion. Figure 2c demonstrates an example of the in vivo fiber targeting of subcortical thalamic regions. Also noteworthy is the bleeding-induced T2-weighted signal drop when the fiber was inserted through the lateral ventricle (Fig. 2c). When a fiber tip first reaches a ventricle, its pushing force causes deformation of the surrounding ependyma, which can induce minor bleeding from the choroid plexus. This observation should raise a note of caution to target deep brain regions. The damage could be reduced by decreasing the insertion speed, which can be accomplished at approximately 20 μm/s with the MgRA (Supplementary Movie 5 and Supplementary Fig. 4).

Fiber optic insertions with customized angles can also be applied with MgRA for the in vivo animal fMRI environment. Figure 2d shows the step-wise fiber tip targeting to the hippocampus and ventral posteromedial nucleus (VPM) of the thalamus by inserting the optical fiber with a 40° angle from the midline. Figure 2e demonstrates the whole brain BOLD fMRI map upon optogenetic activation of either the hippocampus or the VPM, based on the MgRA-driven step-wise fiber tip localization. Thus, the implementation of MgRA in standard opto-fMRI workflows provides flexibility to guide an optical fiber along a certain insertion trajectory, allowing to target different nuclei in a single fMRI experiment, and hence, to study whole brain responses upon deliberate region-specific stimulation.

**Whole brain fMRI with LH optogenetic activation.** The MgRA can be used to target the deep brain nuclei with much higher precision for fiber optic-mediated optogenetic activation than the conventional stereotaxic-based fiber implantation on bench. For example, the LH is a heterogeneous nucleus with highly varied cell types across a few millimeter space in the ventral brain[31]. The MgRA-driven fiber optic positioning provides a reliable and precise targeting scheme for the LH optogenetic activation during fMRI. Figure 3a shows ChR2 expression with the AAV viral vector AVV9.CaMKII.ChR2.eYFP into the LH and the fiber optic trace to target the LH in the histological slice, as well as the MR image showing how the fiber tip coincides with the traced site of viral injection. The whole brain activation pattern upon the LH optogenetic activation is presented in Fig. 3b, showing the blood oxygen level dependent (BOLD) signal along the ascending projection to the basal forebrain from the LH. Figure 3c shows the temporal evolution of the optogenetically evoked BOLD signals in both LH and its projected basal forebrain regions with the mean time courses acquired at different stimulation durations. Figure 3d shows the mean BOLD signal time courses from both nuclei with varied optical light pulse frequencies and pulse widths (whole brain functional patterns at varied pulse width are shown in Supplementary Fig. 5). The BOLD amplitude dependency on the light pulse parameters provides strong evidence for reliable detection of the functional projections from the LH with optogenetic fMRI. It is also noteworthy that MgRA-driven fiber optic implantation ensures highly comparable activation patterns in the LH across different animals (results from 5 individual rats, Fig. 3b), as well as the activation of areas in the basal forebrain including the lateral preoptic area (LPO), medial preoptic area (MPA), and the strial part of the preoptic area (StA) (the co-registered brain atlas to the individual rat functional map, Fig. 3b). Additionally, the evoked calcium and BOLD signals in the barrel cortex (BC) were observed in these animals upon somatosensory whisker stimulation (Supplementary Fig. 6), which indicates a stable physiological state of the animal and therefore validates the biological data acquired from these experiments. These results indicate that MgRA provides high targeting accuracy and effectiveness to target deep brain circuits and produce optogenetically-driven brain activation in a highly reliable manner.

**Step-wise optogenetically driven fMRI and calcium recording.** The flexibility and high precision of MgRA-driven fiber optic targeting was further verified in a series of experiments that combined optogenetic activation with concurrent fMRI and calcium fiber optic recording (Fig. 4a). This multi-modal fMRI

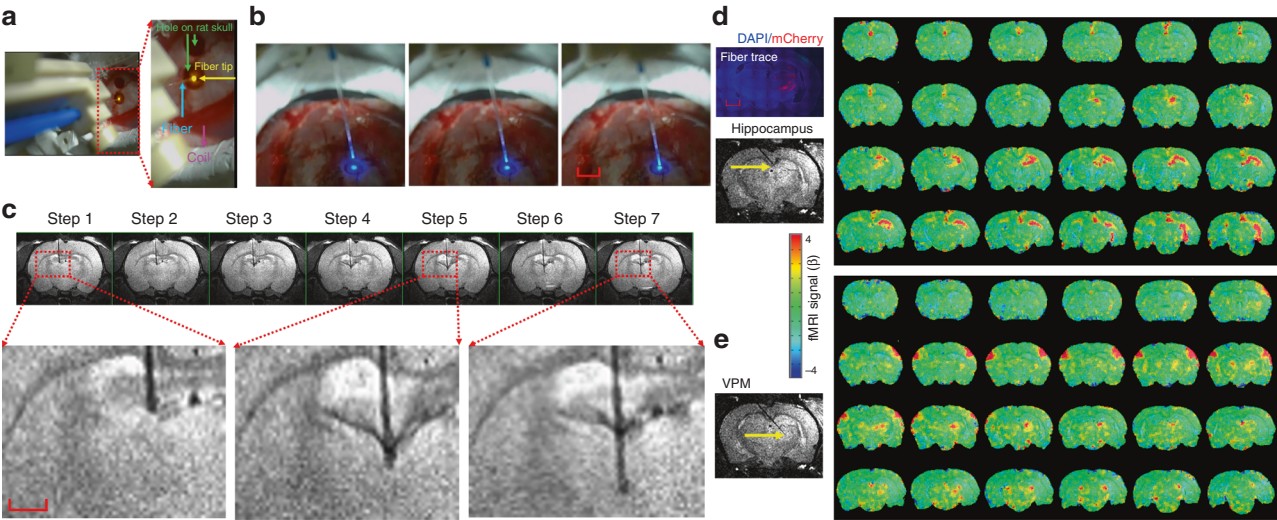

**Fig. 2** Evaluation of MgRA for in vivo studies and brain-wide opto-fMRI patterns in multiple targets. **a** Snapshot of the optical fiber (tip with 589 nm wavelength laser light, yellow arrow) positioned above the burr hole (green arrow) on the skull of an anesthetized rat with the driving piece of the head part. **b** Camera-based fiber optic movement for three steps outside the rat brain. The fiber tip delivers blue laser light (473 nm wavelength). The bright ring-structure above the rat skull is the RF surface coil. Scale bar, 2 mm. **c** Potential collateral damage from the choroid plexus when the fiber was lowered to pass through the lateral ventricles, shown as a dark signal below the hippocampus. The step size was 300 μm. Scale bar, 1 mm. **d** Left, histological image demonstrates ChR2-mCherry expression in most of the thalamus and part of hippocampus. Red, ChR2-mCherry; blue, 4′,6-Diamidino-2-phenylindole (DAPI). Right, the fiber tip targets the hippocampal area and the BOLD fMRI map shows the activated area primarily located in the ipsilateral hippocampal structure. Scale bar, 2 mm. **e** The fiber tip targets the ventral postero-medial (VPM) thalamus and map of BOLD activity was detected in bilateral vibrissal S1 cortex in response to blue light stimulation. (For both (**d**) and (**e**), 3D whole brain EPI: 400 μm isotropic resolution, 1.5 s repetition time; stimulation block design: 8 s on 37 s off; laser pulse: 10 ms, 5 Hz, 3.7 mW/200 μm core diameter of fiber tip)

scheme with MgRA enables real-time feedback at the level of the whole brain (via fMRI) and specifically from the fiber tip (via optical fiber) regarding the activation of the projection structures upon region-specific stimulation. Here, calcium imaging was acquired from the neurons in the BC that received afferents from the subcortical thalamic region by using the calcium reporter GCaMP6[4,5,12]; optogenetic stimulation was performed on the VPM thalamic nuclei, after expression of the light-sensitive protein ChR2 (Fig. 4b–d)[2,3,32]. The recording fiber was directly implanted to record the GCaMP6f-mediated calcium signal in the BC, while the optogenetic activation fiber was controlled by the MgRA inside the scanner with real-time anatomical and functional MRI to track the insertion trajectory. The MgRA guided the fiber tip to deliver the optogenetic activation at multiple sites along the insertion trajectory (Fig. 4e and Supplementary Movie 6). Evoked calcium and BOLD signals from the somatosensory cortex ipsilateral to the targeted thalamic nucleus increased in a stepwise manner as the optical fiber was moved closer to the VPM region, while, after the fiber bypassed the VPM region, BOLD and calcium signal decreased accordingly (Fig. 4f–i). There was a slightly different stepwise fMRI response from the contralateral somatosensory cortex as well (Fig. 4h, i), which has been previously reported with electrical stimulation[33,34]. To further demonstrate the reliability of MgRA, five power levels of light pulses were used to trigger increased BOLD and simultaneous calcium signals (Supplementary Fig. 7). Moreover, by altering the frequency of the light from 0.5 to 5 Hz, we could observe a fully recovered evoked calcium baseline signal at 0.5 Hz and elevated calcium signals from 1 to 3 Hz, while at 5 Hz, the overall plateau amplitude was not further increased (Supplementary Fig. 8). The BOLD signal increased with higher frequency, but not at 5 Hz, which was consistent with the calcium signal dynamics (Supplementary Fig. 8). Results from two additional rats with different or similar insertion trajectories

confirmed the reliability of the stepwise optogenetic activated fMRI and calcium signals acquired using the MgRA (Supplementary Figs. 9 and 10). These experiments further demonstrate the unique capability of the MgRA to specifically target subcortical nuclei, which, combined with cortical recordings in the projection area, allow unequivocal stimulation of the target sites.

**MgRA-driven Mn-injection into CL and LH.** The MgRA can also be used to guide the real-time microinjection with high precision inside the MRI scanner. MnCl₂ solution was used as the MR contrast agent and a modified MPRAGE sequence[35] (Mdeft, ~4 min) was implemented to detect the manganese-enhanced T1-weighted MRI signal[15,36–38]. As shown in Fig. 5a, a hollow core optical fiber[39–41] was used to target the central lateral thalamic nucleus (CL) and Mn solution was delivered in two consecutive steps. The initial stop was introduced to target the corpus callosum with a small dosage of Mn delivery (Fig. 5b), illustrating the real-time guided injection to target the callosal fibers with a few hundred micron thickness. When the fiber tip as located at the CL (position was verified with a T2-weighted MR image (RARE) overlapped with the brain atlas), Mn solution was injected for three times to show dose-dependent signal changes in the T1-weighted Mdeft images acquired before and after Mn injection (Fig. 5b). This result demonstrates the real-time injection capabilities of the MgRA.

Besides the multiple stops along the single trail of injection trajectory, the MgRA can be used to drive multi-trial microinjection, e.g., to the lateral hypothalamic nucleus from the same rat, inside the MRI scanner. As shown in Fig. 5c, the fiber tip was guided to target the LH. The Mdeft images were acquired before and after the injection (3 times, Fig. 5d, e), showing clear effective Mn delivery to the LH. In addition, we continuously acquired the Mdeft images within the first ~1 h following the injection, showing highly robust and confined Mn-enhanced signal of the

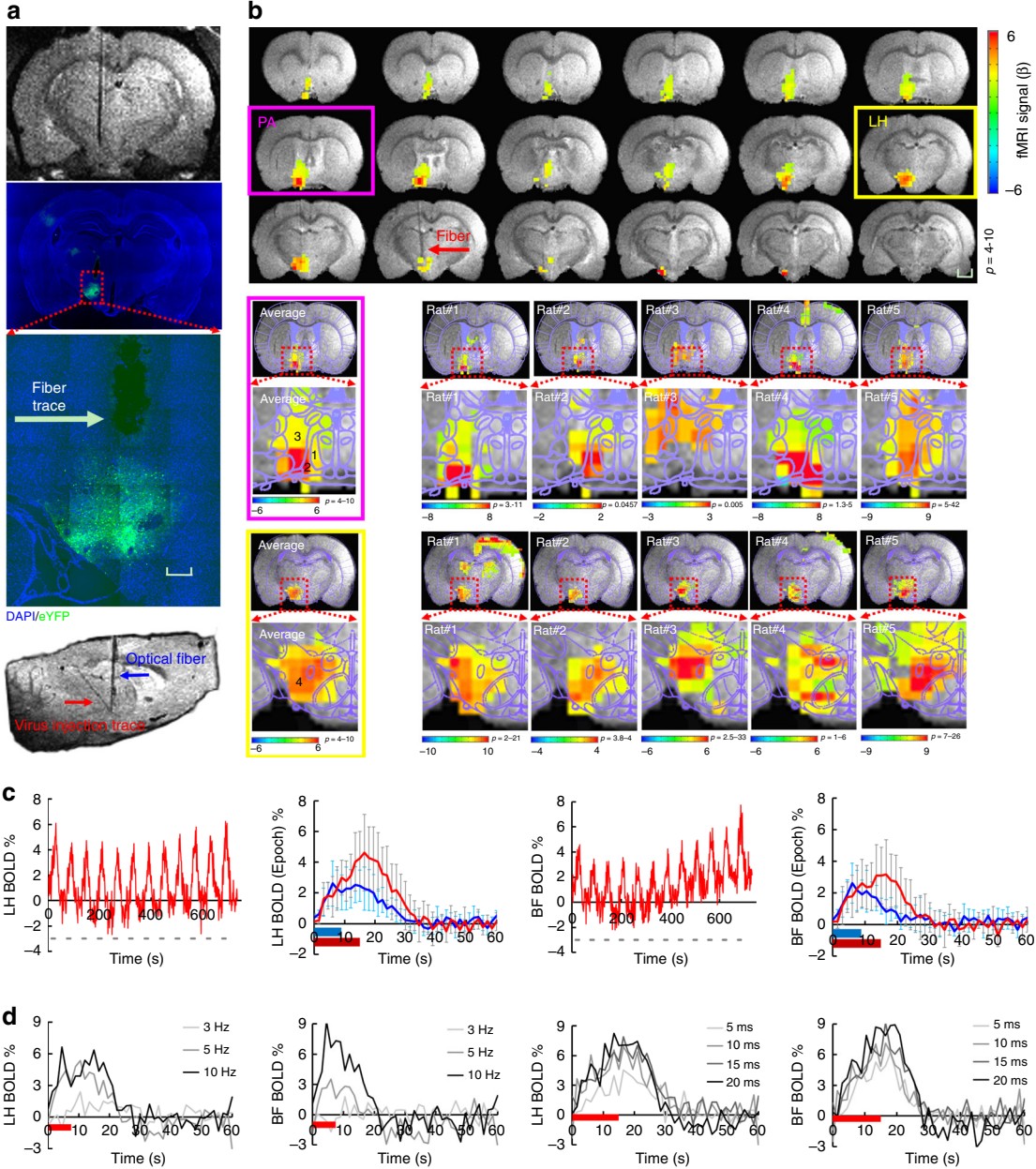

**Fig. 3** MgRA-driven fiber optic targeting of the lateral hypothalamic nuclei with optogenetic fMRI. **a** Top: representative RARE anatomical image used to clarify the optical fiber location driven by MgRA for optical stimulation in LH. Middle: representative wide-field fluorescence image illustrating robust ChR2-eYFP expression focused on LH. Fiber optic insertion trace marked with white arrow. Scale bar, 200 μm. Bottom: sagittal RARE anatomical image showing the fiber optic trace (blue arrow) and virus injection trace (red arrow). **b** Top: average fMRI map of brain-wide activity during optogenetic stimulation of LH neurons at 5 Hz, 20 ms pulse width, 15 s duration. Middle: averaged evoked BOLD map (left) and the same map from 5 individual rats (right) zoomed in on the basal forebrain (BF) showing activation of the lateral preoptic nucleus (LPO) and medial preoptic area (MPA), overlaid with the brain atlas. Bottom: average evoked BOLD map (left) and 5 individual rats (right) in lateral hypothalamic region, overlaid with the brain atlas. GLM-based *t*-statistics in AFNI is used. Scale bar, 2 mm. **c** Average time courses of significantly modulated voxels showing fMRI signal changes within the ipsilateral LH and BF (*n* = 5 animals) upon optogenetic stimulation of block design: 15 s on/45 s off, 12 epoch, 20 ms light pulse, 5 Hz, 18.9 mW. The individual hemodynamic response shows the average BOLD signal upon different stimulation durations (8 s in blue, 15 s in red). Error bars represent mean ± SD across 5 animals. **d** Average stimulation duration-locked time evolution for both LH and BF depicting the frequency-dependent hemodynamic responses at 3, 5, and 10 Hz with 8 s stimulus duration, as well as pulse-width-dependent hemodynamic responses at 5, 10, 15, and 20 ms with 15 s stimulus duration, from one representative rat

targeted regions with limited diffusion (Fig. 5f and Supplementary Movie 7). The MgRA-driven microinjection was reproduced in multiple animals, suggesting a highly robust performance of the MgRA to target deep brain nuclei for injection purposes, as quantified in Fig. 5g. The high spatial specificity of MgRA-driven

microinjection can be used to improve the tract-tracing studies with MEMRI[15,36–38], as well as to optimize the real-time in vivo neuromodulation or molecular MRI by direct intracranial injection of drugs[42–44] and MRI contrast sensors for neuro-transmitters[45–49].

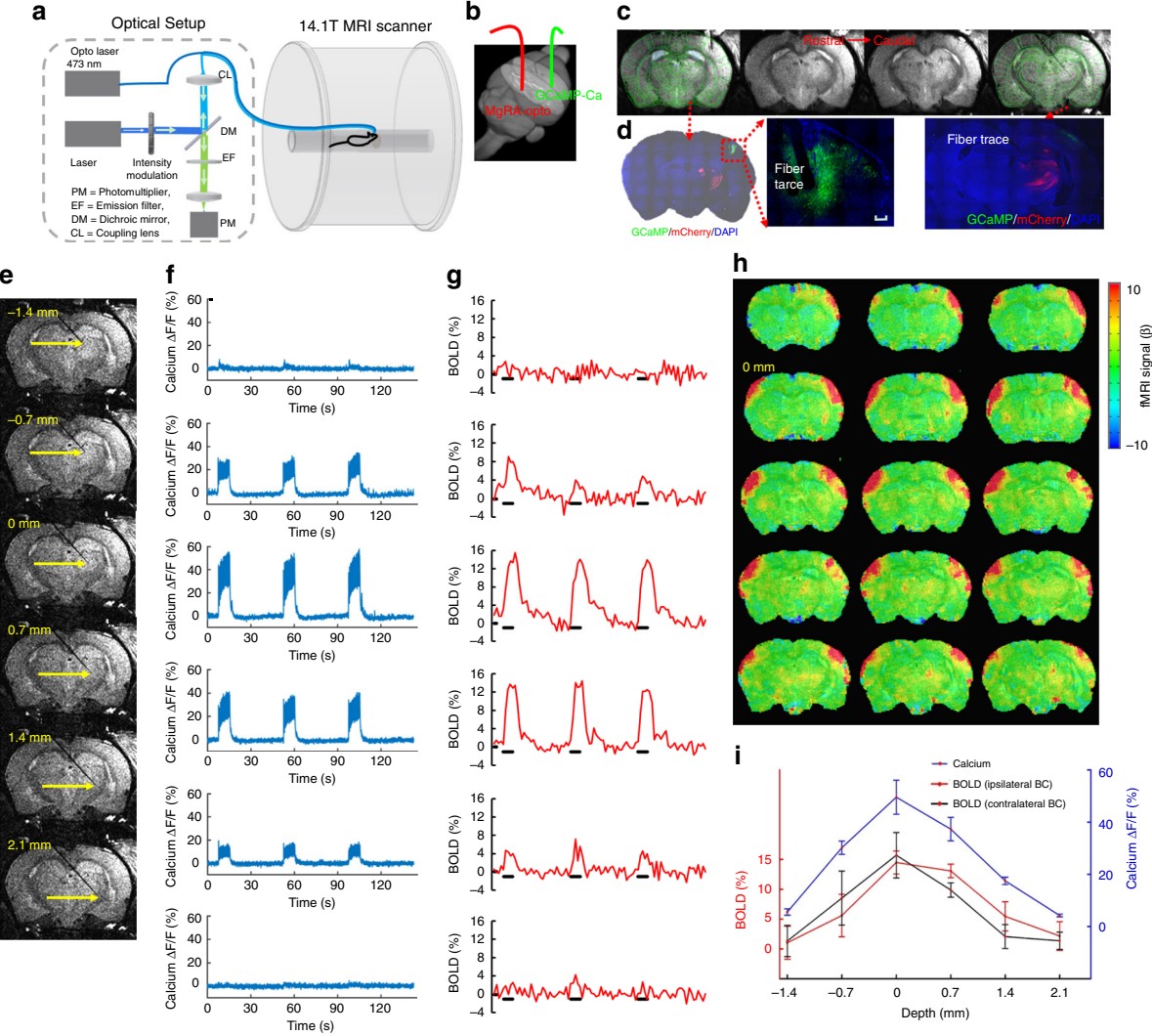

**Fig. 4** MgRA-driven stepwise optogenetic activation of the thalamic nuclei with simultaneous fMRI and neuronal Ca$^{2+}$ recordings. **a** Schematic drawing of the experimental setup to conduct optogenetic fMRI with simultaneous fiber optic calcium recording. The optical setup was placed outside the 14.1 T scanner. Opto Laser: laser for optogenetics. **b** Schematic of the fiber optic insertion inside the rat brain (3D view) with MgRA-controlled optical fiber for optogenetic activation (red) and a second optical fiber for calcium recording (green) in the barrel cortex. **c** The anatomical MRI images confirm the location of the recording fiber and the stimulation fiber targeting the VPM thalamic region. The brain atlas is superimposed on the anatomical image (green). **d** The immunostaining images show the ChR2 expression in the thalamic region (ChR2-mCherry marked in red), as well as the GCaMP6f expression (green) in the vibrissal S1 cortical neurons (BC) with the fiber trace. Scale bar, 200 μm. **e** Anatomical RARE MR images illustrate the fiber tip location at 6 steps, at a step-size of 700 μm. **f** Percentage changes of the evoked calcium signal for 3 epochs upon light stimulation (3 Hz, 10 ms pulse width, 3.7 mW laser power, 8 s on 37 s off block design). **g** Simultaneous BOLD signals for 3 epochs within the ipsilateral somatosensory cortex (see (h)). **h** Evoked BOLD fMRI map when the fiber tip was positioned at 0 mm along the insertion trajectory (zero considered as the position that leads to the peak fMRI and calcium signals). **i** Average amplitudes of the ipsilateral evoked calcium and BOLD signals of both hemispheres as a function of the fiber tip locations. Error bars represent mean ± SD

## Discussion

This work presents an MRI compatible robotic arm as the navigation technique for accurate placement of optical fibers in multimodal fMRI studies in animals using ultra high-field MRI (14.1 T scanner). The MgRA was first developed and improved with a series of phantom tests and was posteriorly evaluated in vivo for deep brain optical fiber placement. MgRA-driven optogenetic activation at subcortical nuclei, e.g., LH and VPM, in a stepwise manner not only demonstrates the high precision of MgRA to target subcortical brain nuclei as deep as 8–9 mm from the skull surface, but also increases the reproducibility of the region-specific optogenetic activation for the whole-brain fMRI mapping in combination with the concurrent fiber optic calcium recordings. Also noted is that the mobility range of the MgRA (10 mm

in the rostral–caudal and medial–lateral directions) is sufficient to reach any brain structure in small animals for optogenetic fMRI and intracellular calcium recording. In addition, the MgRA was applied for real-time microinjection to specific deep brain nuclei, as demonstrated with an Mn-enhanced MRI method, demonstrating its microinjection capabilities for contrast agent or drug delivery with high precision inside the MRI scanner.

The main challenge when targeting deep brain structures is the potential error that appears between the actual and the calculated coordinates due to the variability in bregma location, skull thickness/angles, and potential shift of brain structures within the cranium after dura removal[50–52]. This potential error is particularly problematic when targeting some functional nuclei or neuronal fiber tracts of the rat brain that are less than 2–300 μm in

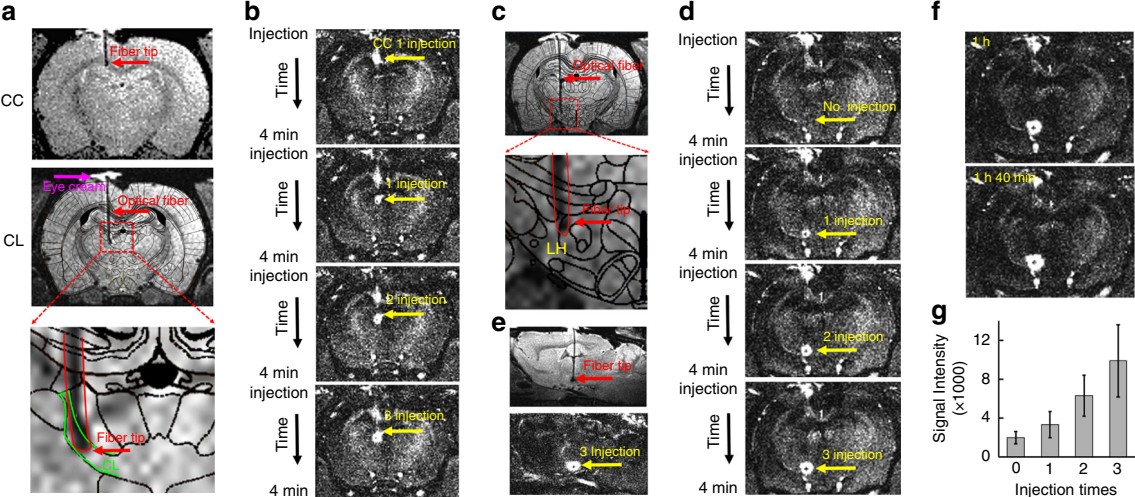

**Fig. 5** MgRA-driven Mn-injection into CL and LH. **a** Top: the representative RARE anatomical image used to clarify the optical fiber location driven by MgRA for Mn injection in CC. Middle: the atlas overlapped RARE images to illustrate the fiber tip location at the CL. Eye cream is covering the craniotomy (magenta arrow). Bottom: enlarged image of fiber location. **b** T1-weighted MPRAGE image (Mdeft) showing enhanced signal from Mn injection site in the CC and CL with dose-dependency. **c** Top: the atlas overlapped RARE images to illustrate the fiber tip location at the LH, Bottom: enlarged image of fiber location. **d** T1-weighted MPRAGE image showing enhanced signal from Mn injection site in the LH with dose-dependency. **e** Sagittal view of RARE anatomical image and MPRAGE image after MnCl$_2$ solution injection. **f** T1-weighted MPRAGE image at 1 h and 1 h 40 min after the injection. **g** The analysis of MEMRI signal at no injection, 1 injection, 2 injection, and 3 injection times, as shown in (**b**, **d**) ($n = 5$ injection points from 3 animals). Error bars represent mean ± SD

one of their dimensions, such as the central thalamic nuclei or corpus callosal fibers[38,53]. This problem can produce high variability when we try to target the deeper brain nuclei, e.g., LH, since longer trajectories are subjected to larger errors[8,14]. In order to optimize the positioning of the optical fiber into precise coordinates of the rat brain, we propose to avoid the atlas-base blind implantation by using a real-time feedback strategy that allows visualization of the whole brain with MRI during fiber insertion. We designed an MRI-compatible robotic arm which allows lowering the optical fiber inside the rat brain with real-time MRI scanning. By combining MRI guidance with the precise control of four stereotactic parameters (radial angle, rostral–caudal, dorsal–ventral, medial–lateral), the MgRA can fine-tune the fiber positioning to conduct highly reproducible and stepwise optogenetic fMRI studies.

The number of applications for robotic arms in animal research is considerably increased as a result of their potential combination with MRI. Examples include an MR image-guided mini-DBS system for BOLD activation during subthalamic nucleus DBS in nonhuman primates in a 3 T scanner[26], an angle positioning system to increase the image signal intensity of fibrous microstructure in a 9.4 T 12 cm-bore scanner[27], an integrated system, driven by piezoelectric actuators, for auto-tuning of a multichannel transceiver array at 7 T[28] or MRI-compatible systems for focused ultrasound experiments in rodents in 3 T scanners[54,55]. Here, we developed a stepper motor-controlled compacted MgRA system in a 14.1 T horizontal MRI scanner with built-in MRI compatible cameras and RF surface coils to drive fiber optic insertion for optogenetic fMRI studies with concurrent intracellular calcium recordings. To our knowledge, this is the first time to combine the multi-modal fMRI neuroimaging platform with the MRI-guided robotic controlling system for in vivo rodent brain functional mapping.

There are two key advantages that need to be highlighted from the mechanical design of the MgRA system. In high-field MRI scanners, the open space inside the magnetic bore above the animal brain is usually less than 3–4 cm, which significantly limits the kinematic design options for mechanical movement. Also, the

ultra-high field (>11.7 T) also limits the commercially available motor supplies that avoid the electromagnetic interference with the MR scanning. We designed the MgRA head-probe based on an Archimedean spiral mechanism to achieve high precision and accuracy to maneuver the optical fiber insertion at less than 50 μm step-size along the dorsal–ventral axis (Fig. 1g, Supplementary Fig. 11, and Supplementary Movie 4). This head-probe is controlled by a synchronous belt drive, which can carry up to 4 degree-of-freedom movements inside the horizontal bore of the 14.1 T MRI scanner (Supplementary Fig. 12), and only occupies 1.5–2 cm space.

To deal with the MRI compatibility, in addition to hydraulic[56,57] or pneumatic[27,58–60] actuators, other types such as ultrasonic or piezoelectric motors, which have been the favorite so far due to their non-magnetic core, short response time and small size[61,62], could have been utilized. However, no commercially available piezo motors are available for the 14.1 T MRI scanner and it has been recently shown that piezo motors could induce geometric distortions in MR images even at a lower magnetic field strength[63,64]. Also, different MRI sequences could have effects on the behavior of ultrasonic motors[65]. To address the compatibility issue, remotely actuated MR-compatible manipulators were implemented using drive shafts, belts, chain drive, and linkages to transfer the motion to the distant actuated points[66–68]. We have applied the long robotic arm to allow us to apply the regular stepper motor to control the optical fiber insertion. As shown in the Supplementary Movies 2–5, the mechanical control of the optical fiber insertion remains highly precise and reliable in both ex vivo and in vivo tests. Our MgRA design not only provides a highly robust mechanical controlling system, but also solves the MRI compatibility issue with a reliable and economically affordable solution. We will further optimize our MgRA system by shortening the robotic arm and implementing the piezo motors with a safe distance to avoid electromagnetic interference.

Besides fulfilling the role of accurately placing the fiber tip at the desired coordinates, the MgRA provides a flexible platform (Fig. 1g) to identify, de novo, the ideal targets for deep brain

stimulation in pre-clinical studies. This could be easily investigated with the MgRA by moving the stimulating fiber and running opto-fMRI at different locations in one single study, particularly for "hypothesis-free" brain activity mapping studies. This application will be critical to optimize and specify the ideal subcortical targets aiming at controlling pathological tremor or searching for more reliable treatment for depression in animal models[69–71]. Importantly, certain effects inherent in the insertion of electrodes or optical fiber into the brain can be visualized and avoided using the MgRA strategy. One example is the case of the potential collateral damage to the choroid plexus (Fig. 2c) or other blood vessels, which could be well monitored by real-time imaging and avoided by changing the trajectory of the fiber. This is a particularly relevant feature of the MgRA, as it contributes to the maintenance of certain integrity of the surrounding tissue, which is beyond the capabilities of the standard implantation techniques with stereotactic devices and is crucial for potentially translational studies, as raised in a report showing MRI-guided cell transplantation into the brain[72].

Several limitations pertaining to the first version of the MgRA should be considered when interpreting the results of this study and for future optimization of the MgRA in high field MRI scanner for animal imaging. Firstly, the angle/direction of the optical fiber cannot be changed once that it has been placed inside the brain parenchyma, as this would lead to excessive tissue damage and/or bleeding. Instead, in case needed, the optical fiber should be withdrawn and reinserted; thus it is crucial to improve the algorithm to calculate the trajectory based on the location of the optical fiber tip in the agarose covering the craniotomy outside of the brain parenchyma. Secondly, it is noteworthy that, because of the long arm to keep the stepper motors work properly outside of the MRI scanner, the most precise movement occurs along the ventral–dorsal direction (Fig. 1g and Supplementary Movie 4). It will be an important step forward to implement the piezo motors with a safe distance to avoid electromagnetic interference, which would allow to dramatically shorten the robotic arm and, consequently, to optimize of the precision in all the axis. Thirdly, although we acquired the 3D anatomical images of the rat brain, the major registration procedure between atlas and MRI images is still based on a 2D registration algorithm, which is applied to control the fiber tip movement along the dorsal–ventral direction. In the future development, we will provide a real-time 3D registration system to take advantage of the full motor control movement capability of the MgRA system to achieve a fully automatic performance. Lastly, the precision measurement of the MgRA can be directly evaluated based on the real-time anatomical MRI images. However, the best resolution acquired so far in our MRI scanner is $50 \times 50\,\mu m$ in-plane. The MRI spatial resolution is much lower than the mechanistic movement precision provided by the MgRA system. For future piezo-based micron-resolution motor control system, the implementation of an optical encoder inside the ultra-high magnetic field will be needed for the close-loop feedback.

In summary, the real-time MRI-guidance in a robotic controlling system is verified and practiced for the optical fiber brain intervention in animals using the high field MRI scanner (>14 T). This MgRA positioning system serves as a key component for the future multi-modal fMRI platform merging concurrent fMRI with optogenetics, fiber optic-mediated optical imaging, micro-injection, and even electrophysiological recordings. The high flexibility and precision of MgRA to target the deep brain nuclei with neural circuit-specificity expands the brain functional mapping studies from the cellular levels, to the neural circuit levels, and eventually to the systems' levels in combination with behavioral tests in animals.

## Methods

**MgRA system.** The MgRA was manufactured by the Fine Mechanical and Electrical Workshop in the Max Planck Institute for Biological Cybernetics, Tuebingen, Germany. This system consists of a positioning module, the head of the MgRA, and a custom-designed user interface. The positioning module (back part) accommodates the stepper motors (ST4118D1804-B, Nanotec, Germany) to fulfill the optical fiber movement with multi-degree of freedom, and the head of the MgRA (front part) includes the driving pieces, cameras, and a customized rat holder (Fig. 1b). The coupling of the actuators (back part) to the matching toothed pulley in the head was achieved by a synchronous belt (Optibelt OMEGA 3M, OPTI-BELT, Germany) drive in a form-fit manner. The driving pieces with Archimedean spiral mechanism were manufactured manually or with a 3D printer (Form 2, Formlabs, Germany). The detailed design and components are shown in Fig. 1a–c, Supplementary Figs. 11–13, with a table of all components and the European patent as the following link: https://patentscope.wipo.int/search/en/detail.jsf?docId=EP215319263&tab=PCTDESCRIPTION&maxRec=1000. The movements include three dimensions like conventional stereotactic devices, as well as pitch and yaw (manually). With MRI-compatible cameras (RS-OV7949-1818, Conrad Electronic, Germany), the user can watch the fiber insertion in real time, while the robot is executing a maneuver. If any movement needs to be modified, the user can start, stop, change, or resume the fiber movement at any time from the user-interface. Most of the other components are constructed from fully MRI-compatible materials like plastic, carbon fiber, and a minimal amount of non-ferrous metals like brass and anodized aluminum to avoid eddy currents and deterioration of magnetic field homogeneity. The MRI-compatible arm including the head part and aluminum holder were placed inside the MRI scanner room. Digital components including stepper motors (ST4118D1804-B, Nanotec, Germany), the motor controller (SMCI33-1, Nanotec, Germany) and motor power supply (NTS-24V-40A, Nanotec, Germany), were placed outside the scanner room (Supplementary Fig. 1).

**Viral injection.** The study was performed in accordance with the German Animal Welfare Act (TierSchG) and Animal Welfare Laboratory Animal Ordinance (TierSchVersV). This is in full compliance with the guidelines of the EU Directive on the protection of animals used for scientific purposes (2010/63/EU). The study was reviewed by the ethics commission (§15 TierSchG) and approved by the state authority (Regierungspräsidium, Tübingen, Baden-Württemberg, Germany). A total of 21 male Sprague–Dawley rats were used in this study.

Intracerebral viral injection was performed in 3–4-week-old male Sprague–Dawley rats to express the viral vectors containing the calcium-sensitive protein (GCaMP for calcium recording) or the light-sensitive protein channelrhodopsin-2 (ChR2 for optogenetics) in neurons. The construct AAV5.Syn.GCaMP6f.WPRE.SV40 (2.818e13 genome copies per milliliter) was used to express GCaMP in the BC and the constructs AAV9.CAG.hChR2(H134R)-mCherry.WPRE.SV40 (2.918e13 genome copies per milliliter) and AAV9.CaMKII.hChR2 (E123A)-eYFP.WPRE.hGH (1.19e13 genome copies per milliliter) were used to express ChR2 in the thalamus and LH, respectively. Rats were anesthetized with 1.5–2% isoflurane via nose cone and placed on a stereotaxic frame, an incision was made on the scalp and the skull was exposed. Craniotomies were performed with a pneumatic drill so as to cause minimal damage to cortical tissue. For optogenetics, a volume of 0.6–1 μL was injected using a 10 μL syringe and 33-gauge needle. The injection rate was controlled by an infusion pump (Pump 11 Elite, Harvard Apparatus, USA). The stereotaxic coordinates of the injections were 2.5 mm posterior to Bregma, 5.0 mm lateral to the midline, 0.8–1.4 mm below the cortical surface to target the BC; 2.6–2.7 mm posterior to Bregma, 2.8 mm lateral to the midline, 5.5–6.0 mm below the cortical surface for the ventral posterior medial nucleus of thalamus (VPM); and 2.75–2.85 mm posterior to Bregma, 1.1 mm lateral to the midline, 7.5–7.9 mm below the cortical surface for LH. After injection, the needle was left in place for approximately 5 min before being slowly withdrawn. The craniotomies were sealed with the bone wax and the skin around the wound was sutured. Rats were subcutaneously injected with antibiotic and painkiller for 3 consecutive days to prevent bacterial infections and relieve postoperative pain.

**Animal preparation for fMRI.** Anesthesia was first induced in the animal with 5% isoflurane in chamber. The anesthetized rat was intubated using a tracheal tube and a mechanical ventilator (SAR-830, CWE, USA) was used to ventilate animals throughout the whole experiment. Femoral arterial and venous catheterization was performed with polyethylene tubing for blood sampling, drug administration, and constant blood pressure measurements. After the surgery, isoflurane was switched off and a bolus of the anesthetic alpha-chloralose (80 mg/kg) was infused intravenously. A mixture of Alpha-Chloralose (26.5 mg/kg/h) and pancuronium (2 mg/kg/h) was constantly infused to maintain the anesthesia/keep the animal anesthetized and reduce motion artifacts.

**Fiber optic implantation and optogenetic stimulation.** Before transferring the animal to the MRI scanner, 2 craniotomies were performed. Briefly, the animal was placed on a stereotaxic frame, the scalp was opened and two ~1.5 mm diameter burr holes were drilled on the skull. The dura was carefully removed and an optical fiber with 200 μm core diameter (FT200EMT, Thorlabs, Germany) was inserted

into the BC, at coordinates: 2.75–3.3 mm posterior to Bregma, 5.0 mm lateral to the midline, 1.2–1.4 mm below the cortical surface. An adhesive gel was used to secure the calcium recording fiber to the skull. The craniotomy for optogenetics (in VPM or LH) was covered by agarose gel for robotic arm-driven fiber insertion inside the MRI scanner. Toothpaste was applied within the ears to minimize MR susceptibility artifacts for the whole brain fMRI mapping. The eyes of the rats were covered to prevent stimulation of the visual system during the light-driven fMRI.

For optogenetic stimulation, square pulses of blue light (473 nm) were delivered using a laser (MBL-III, CNI, China) connected to the 200 μm core optical fiber (FT200EMT, Thorlabs, Germany) and controlled by Master 9 (Master-9, A.M.P.I., Israel). The light intensity was tested before each experiment, and was calibrated with a power meter (PM20A, Thorlabs, Germany) to emit 0.6–40 mW from the tip of the optical fiber for LH and thalamus. The power levels used for light-driven fMRI studies did not induce pseudo-BOLD signal due to heating effects, by testing in regions of interest both with and without ChR2 expression.

**Immunohistochemistry.** To verify the phenotype of the transfected cells, opsin localization, and optical fiber placement, perfused rat brains were fixed overnight in 4% paraformaldehyde and then equilibrated in 15 and 30% sucrose in 0.1 M PBS at 4 °C. 30 μm-thick coronal sections were cut on a cryotome (CM3050S, Leica, Germany). Free-floating sections were washed in PBS, mounted on microscope slides, and incubated with DAPI (VectaShield, Vector Laboratories, USA) for 30 min at room temperature. Wide-field fluorescent images were acquired using a microscope (Zeiss, Germany) for assessment of GCaMP expression in BC, ChR2 in LH and VPM. Digital images were minimally processed using ImageJ to enhance brightness and contrast for visualization purposes.

**Optical setup.** An OBIS laser was used as excitation light source (OBIS 488LS, Coherent, Germany) with a heat sink to enable laser operation throughout the entire specified temperature range from 10 to 40 °C. The light passed through a continuously variable neutral density filter (NDC-50C-2M-B, Thorlabs, Germany) and was reflected on a dichroic beam splitter (F48-487, AHF analysentechnik AG, Germany). The beam was collected into an AR coated achromatic lens (AC254-030-A, Thorlabs, Germany) fixed on a threaded flexure stage (HCS013, Thorlabs, Germany) mounted on an extension platform (AMA009/M, Thorlabs, Germany) of a fiber launch system (MAX350D/M, Thorlabs, Germany). The laser beam was injected into the fiber and propagated to the tip. The emitted fluorescence was collected through the fiber tip, propagated back and collimated by the achromatic lens, passed through the dichroic beam splitter and filtered by a band-pass filter (ET525/50M, Chroma, USA) and focused by an AR coated achromatic lens (AC254-030-A, Thorlabs, Germany). A silicon photomultiplier module (MiniSM 10035, SensL, Germany) was applied to detect the emitted fluorescence. The entire optical system was enclosed in a light isolator box. The photomultiplier output was amplified (gain = 100) by a voltage amplifier (DLPVA-100-BLN-S, Femto, Germany), digitized and detected by Biopac system (MP150 System, BIOPAC Systems, USA).

**MRI image acquisition.** All images were acquired with a 14.1 T/26 cm horizontal bore magnet interfaced to an Avance III console and equipped with a 12 cm gradient set capable of providing 100 G/cm over a time of 150 μs. A transceiver single-loop surface coil with an inner diameter of 22 mm was placed directly over the rat head to acquire anatomical and fMRI images. Magnetic field homogeneity was optimized first by global shimming for anatomical images and followed by FASTMAP shimming protocol for EPI sequence.

Anatomical images were acquired for approximate fiber location using 3D FLASH MRA sequence with the following parameters: repetition time, 20 ms; echo time, 2.82 ms; FOV: 2.28 cm × 2.28 cm × 2.28 cm, matrix = 114 × 114 × 114, spatial resolution = 0.2 mm × 0.2 mm × 0.2 mm. A high-resolution RARE sequence was used accurately identify the optical fiber in the coronal plane, with the following parameters: repetition time, 1200 ms; echo time, 7 ms; FOV: 1.92 cm × 1.68 cm, matrix = 128 × 112, resolution = 0.15 mm × 0.15 mm, slice thickness = 0.5 mm, RARE factor = 8, averages = 16.

Higher resolution (50 μm) RARE sequence, specifically for Fig. 1f, g, to accurately identify the optical fiber in the coronal plane, with the following parameters: repetition time, 1500 ms; echo time, 11.0428 ms; FOV: 1.92 cm × 1.56 cm, matrix = 384 × 312, resolution = 50 μm × 50 μm, slice thickness = 0.75 mm, RARE factor = 6, averages = 6.

For Mn injections and Mn tracing studies, rats received 150 nL of 5 mM MnCl$_2$ (MnCl$_2$, Sigma-Aldrich, Germany) solution for three times delivered by a hollow core photonic crystal fiber (diameter: ~240 μm)[39–41], manufactured by the Division of Photonic Crystal Fibre Science at Max-Planck Institute for the Science of Light, Erlangen, Germany. A magnetization prepared rapid gradient echo (MP-RAGE) sequence[35] was used. Eight coronal slices with FOV = 1.92 × 1.92 cm, matrix 128 × 128, thickness = 0.7 mm, (TR = 4000 ms, echo TR/TE = 15/1.7 ms, TI = 1000 ms, number of segments = 4, averages = 2), were used to cover the area of interest at 150 μm in-plane resolution with total imaging time 4 min 16 s. A same field of view T2-weighted RARE sequence was used with the following parameters: repetition time, 3000 ms; echo time, 8.3333 ms; FOV: 1.92 cm × 1.92 cm, matrix = 128 × 128,

resolution = 150 μm × 150 μm, slice thickness = 0.7 mm, RARE factor = 6, averages = 4.

**Functional MRI acquisition.** Adjustments to echo spacing and symmetry, and B$_0$ compensation were set up first. Functional images were acquired with a 3D gradient-echo EPI sequence with the following parameters: echo time 12.5 ms, repetition time 1.5 s, FOV 1.92 cm × 1.92 cm × 1.92 cm, 48 × 48 × 48 matrix size, spatial resolution = 0.4 mm × 0.4 mm × 0.4 mm. To reach steady state 10 dummy scans were used. For anatomical reference, the RARE sequence was applied to acquire 48 coronal slices with the same geometry of the fMRI images.

For fMRI studies, needle electrodes were placed on the forepaw or whisker pads of the rats, and electric pulses (333 μs duration at 1.5 mA repeated at 3 Hz for 4 s) were first used as stimulation to serve as positive control for the evoked BOLD signal. Once that reliable fMRI signals were observed in response to electrical stimulation, optical stimulation was performed. An optical fiber of 200 μm core diameter (FT200EMT, Thorlabs, Germany) was connected to a 473 nm laser source (MBL-III, CNI, China) using a built-in FC/PC coupler to deliver blue light pulses at 3-10 Hz, 5-20 ms pulse width with different durations. To reach steady state 10 dummy scans were used and followed by 10 pre-stimulation scans, 5 scans during stimulation, and 25 inter-stimulation scans for 10 epochs and 5 scans during stimulation and 35 inter-stimulation scans for 12 epochs for thalamus and LH, respectively. The stimulation control was established using the BIOPAC system (MP150 System, BIOPAC Systems, USA) and Master 9 (Master-9, A.M.P.I., Israel).

**Data analysis.** For evoked fMRI analysis, EPI images were first aligned to anatomical images acquired in the same orientation with the same geometry. The anatomical MRI images were registered to a template across animals, as well as EPI datasets. The baseline signal of EPI images was normalized to 100 for statistical analysis of the multiple runs of EPI time courses. The hemodynamic response function (HRF) used the block function of the linear program 3dDeconvolve in AFNI. BLOCK ($L$, 1) is a convolution of a square wave of duration $L$, makes a peak amplitude of block response = 1, with the $g(t) = t^4 e^{-t}/[4^4 e^{-4}]$ (peak value = 1). The HRF model is defined as follows:

$$\mathrm{HRF}(t) = \mathrm{int}(g(t-s), s = 0..\min(t, L))$$

In this case, each beta weight represents the peak height of the corresponding BLOCK curve for that class, i.e., the beta weight is the magnitude of the response to the entire stimulus block.

The fiber optical neuronal calcium signals were low-pass filtered at 100 Hz using zero-phase shift digital filtering. The relative percentage change of fluorescence ($\Delta F/F$) was defined as $(F - F_0)/F_0$, where $F_0$ is the baseline, that is to say, the average fluorescent signal in a 2 s pre-stimulation window. The amplitudes of the neuronal fluorescent signal in response to 4 s optogenetic stimulus (Fig. 4f) were calculated as the average of difference in $\Delta F/F$ in a time window 300 ms after stimulus. Error bars in Figs. 3c, 4i, 5g, and Supplementary Fig. 10f represent standard deviation.

**Reporting summary.** Further information on research design is available in the Nature Research Reporting Summary linked to this article.

## Data availability

The raw data can be provided upon email request to the corresponding author. Excel files containing raw data and each quantitative plot included in the main figures can be found in the Source Data File. For the design of the robotic arm, detailed information can be directly downloaded through the official link of World Intellectual Property Organization (WIPO): https://patentscope.wipo.int/search/en/detail.jsf?docId=EP215319263&tab=NATIONALBIBLIO&maxRec=1000.

## Code availability

The Analysis of Functional NeuroImages software (AFNI, NIH, USA) and Matlab (MATLAB, MathWorks, USA) were used to process the fMRI and simultaneously acquired calcium signals, respectively. The relevant source codes can be downloaded through https://afni.nimh.nih.gov/afni/. The related image processing codes can be provided upon direct email request to the corresponding author.

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

## Acknowledgements

The authors thank Mr. S. Yu for building up the first prototype of the robotic arm and Fine Mechanic and Electronic Workshop at MPI for Biological Cybernetics for MgRA system automation. The financial support of the Max-Planck-Society, the Sino-Germany joint grant by DFG (YU215/3-1645423), and the China Scholarship Council (Ph.D. fellowship to Y. Chen) are gratefully acknowledged. We thank the collaborative support from Dr. G.A. Johnson to provide the original 3D MRI/DTI dataset for image processing and Dr. G. Paxinos for the support of brain atlas (the permission was issued by Elsevier). The authors thank Dr. N. Avdievitch, Ms. H. Schulz, Mr. F. Sobczak, Mr. J.K. Schlüsener, and Mr. H. Huang for technical support, Dr. P. Douay, Dr. E. Weiler, Ms. S. Fischer, and Mrs. M. Pitscheider for animal support, the AFNI team for the software support, the Genetically-Encoded Neuronal Indicator and Effector (GENIE) Program and the Janelia Farm Research Campus for kindly providing viral plasmids.

## Author contributions

X.Y. designed and supervised the research. Y.C., X.Y., P.P.-R., and X.C. performed animal experiments. Y.C., X.Y., and P.P.-R. acquired data. Y.C. analyzed data. X.C. and M.H.F. provided key technical support. X.Y. and Y.C. wrote the manuscript.

## Additional information

**Competing interests:** X.Y. and Y.C. are co-authors of a patent that describes the mechanical design of the MRI-guided robotic arm (EP3315064). The other authors declare no competing interests.

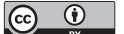

