## [Peer Review File · Nature Communications]

Reviewers' comments:

Reviewer #1 (Remarks to the Author):

In this paper, Chen and colleagues report on a novel MRI-guided robotic arm for activity-guided placement of an optic fiber for optogenetics and optic-fiber-based calcium recordings. The optic fiber is being placed on the robotic actuator, which can be moved in 4 dimensions: three spatial dimensions and angle of insertion. Of particular importance is the full MRI compatibility, which enables continuous fMRI readout during optogenetic manipulation / calcium recordings. The authors provide high-quality data on the feasibility of their approach in terms of controlling the positioning of the fiber for targeting deep brain structures. I fully agree with the assertion of the authors, that their approach represents a very important step forward, we need to tailor the position of optogenetic actuation to the particular, individual functional cytoarchitecture, particularly when targeting deeper brain regions, or for a "hypothesis-free" brain activity mapping. This work represents a mayor advance, I particularly liked the well-controlled and slow advancement of the fiber, which can drastically reduce the probability of intracerebral bleedings when passing the ventricles. The manuscript is overall well written, the figures nicely illustrate the data. However, I have one significant concern, which in my view needs to be conclusively addressed by the authors:

Mayor concern:

1. The authors make a strong and valid claim with regard to the importance of the activity-guided placement of the fiber for optogenetic actuation /calcium recordings. Yet, circuit / region specific optogenetic actuation and / or optical recordings critically also require the similarly precise injection of viral vectors encoding these actuators / indicators. The authors do not at all even mention this critical roadblock. In my view, this MgRA methodology will not lead to a mayor advance, unless paired with a similar approach for activity-guided injection. If the injection of the virus will not be guided by a similar approach, how can the authors state, that the success rate and precision of optogenetics / opto fMRI will be drastically improved? This is particular the case for the cell and region-specific expression of optogenetic actuators in deep regions (as stated for the placement of the fiber (lines 152, but the same holds true for the injection!): the availability of strong cell specific promoters is limited, so we still need a very precise injection scheme. The authors need to address this mayor shortcoming, this paper would be much stronger, if they included an injection capability as well.

Minor comments:

1. Abstract: "intracellular Ca recordings..." this is misleading, as the readers will mistake this term with truly intracellular recordings such as whole-cell patch clamp. The authors conducted somatic calcium recordings
2. Introduction, line 44/45: This general statement, that optic-fiber-based recordings are comparable to electrophysiology recordings is just not true. There are important differences: optical recordings of somatic calcium reflect rather suprathreshold spiking activity, while LFP recordings, which are typically combined with fMRI are dominated by subthreshold, are dominated by synaptic activity. The authors need to be more precise.
3. While I fully support the notion, that the four-dimensional manipulation is a great feature, the authors need to acknowledge / discuss the limitation that it is practically impossible to change the angle / the direction of the fiber once inserted into brain parenchyma, as this will lead to excessive tissue damage and / or bleeding. This needs to be critically discussed.

Reviewer #2 (Remarks to the Author):

This paper describes design, construction and evaluation of a MR-compatible robotic arm to position optic fibers in the rat brain for optogenetic manipulation during fMRI measurements. With the described setup fiber implantations can be monitored by real-time MRI. The authors argue that their apparatus allows for more precise positioning of the fiber than conventional stereotactic

implantations, that fiber position can be corrected or altered during the experiment, and that injuries, potentially causing artefacts, can be observed before starting the actual measurement. Optogenetic methods have recently found widespread application in many fields of neuroscience. In particular the combination with fMRI and optical readout methods has shown great potential for extending our understanding of brain wide and specific neural responses. Therefore, the topic of this paper is of high general interest. Although the direct reproducibility of this work (ie copying the robotic arm) appears not easily feasible for other labs, this paper provides important data, how to improve procedures and data quality in optogenetic experiments, and may thus serve to define new standards. The authors provide data of outstanding quality in terms of (implantation) precision, image resolution and sensitivity of both BOLD and calcium signal. The data fully support the claim that the proposed procedure is superior to conventional stereotactic implantation. Also the gain in data quality and signal level by placing the fiber under MR guidance is impressively illustrated. Further, this paper provides data on neuronal and hemodynamic network response to optogenetic stimulation, which is still not too abundant in the literature.

The paper is well structured and written – it was a pleasure to read.

A few minor issues remain, before this paper may be accepted for publication:

1. Not all components of the robotic arm are described in detail (eg the stepper motor, camera,...). It might be helpful to add a list of all components in the Supplement.
2. Naming of products and companies is incoherent. This should be in a unified format, eg (product, company, country)
3. Line 345: specify: "sealed with bone"
4. Line 360: which adhesive gel was used
5. Line 397-402: Only anatomical MRI with resolution of 150 μm is described. What about the high resolution images in Fig. 1?
6. Line 427 and following unnumbered line: In both equations t has been replaced by 4. Please correct.

I noticed two extra points, which do not appear relevant for the message of the paper. Yet, I am curious to have the authors' comments:

7. In Fig 4, Fig S7, and Fig S8 calcium signal is almost 100 %. Why is this so much higher than in the other measurements?
8. In Fig S8b (3 Hz) and Fig S10d (lower 3 positions) every other calcium response is suppressed/reduced. Similar phenomena are also present in other publications. Yet, I am not aware of an explanation for this variation. Can the authors comment on that.

In summary, I consider this as an excellent piece of work, which should be published after these minor issues have been addressed.

Cornelius Faber, WWU Münster, Germany

Reviewer #3 (Remarks to the Author):

This review is specifically focused on the innovation of the MRI- guided robotic arm (MrGA).

While the authors clearly have a unique application and there is certainly great benefit to an MRI-compatible robotic arm, the device presented does not have the novelty the authors indicate and the review of related literature is completely missing. If the focus of the paper was on what is done with the laser delivery, then this may not be a problem. But, the authors propose the MrGA as a key novel aspect and this is just not true.

Specifically, in line 69 of p.4 the authors state: "there is currently no MRI-compatible, automatic robotic control system to assist fiber optic insertion for multi-modal fMRI platforms." This is simply not true, in fact there is a whole host of research and devices (some in clinical and preclinical trials) for MRI-compatible robotic devices for stereotactic neurosurgery.

Two examples of research systems in preclinical trials are:

<https://www.worldscientific.com/doi/abs/10.1142/S2424905X18500034>

<https://academic.oup.com/neurosurgery/advance-article-abstract/doi/10.1093/neuros/nyy266/5037802>

There are also clinical systems for stereotactic neurosurgery (manual and robotic):

<https://www.mriinterventions.com/clearpoint/clearpoint-overview>

<https://www.monteris.com/neuroplate-system/>

In the rehabilitation robotics community, robots have been used in MRI w/ fMRI such as:

<https://ieeexplore.ieee.org/document/1618680>

Some of these systems are for delivering ultrasonic or fiberoptic probes, typically for thermal ablation or temperature monitoring. They can be used with multi-modal MRI including typical anatomical imaging sequences, thermal imaging, functional imaging, etc.

While these may not be specific to the author's specific application, the authors should compare the novelty of their approach and frame their contributions appropriately (which appears primarily to fit inside a small bore ultra high field scanner).

With regard to the robotic device, the description could use more detail. In particular, it is not clear how the motion is measured. Stepper motors are inherently likely to slip and miss steps, especially if unpowered (which would presumably be the case during as much of the imaging time as possible to minimize electrical noise and vibration). Have the investigators looked into using encoders or another measurement approaches? Are there issues with belt stretch, backlash, slippages that affect the very high stated accuracy (not accuracy study was shown). Was there an assessment of the affect of the motors (electrical noise, vibration, etc) on image quality? There also does not seem to be any discussion of registration to know where the coordinates of the robot arm are relative to the imaging system (and the subject).

In summary, I think the authors need to do a much better job of framing the novel contributions of the work. What is presented is strangely in the discussion at the end of the paper and not up front in the introductions and/or the description of the device. What they are doing with the probe is very interesting, but the robotic devices does not represent any real novel contribution to science and is in fact much less capable than many existing research platforms in the neurosurgery space (which the authors make no attempt to review). Just to clarify, please note that is not to say it is not a well-designed device or that is is not of great utility.

Responses to reviewers

Responses to reviewer #1:

We thank for the very insightful and positive comments from reviewer #1. The reviewer's comments are highly resonant with our original motivation to develop the MgRA system for multi-modality fMRI, including optogenetics. We agree with the reviewer that a crucial microinjection capability of MgRA is missing in this manuscript. Here, we have included new experiments and image analysis to demonstrate the real-time guided microinjection to specific deep brain nuclei using a Mn-enhanced MRI method.

In this paper, Chen and colleagues report on a novel MRI-guided robotic arm for activity-guided placement of an optic fiber for optogenetics and optic-fiber-based calcium recordings. The optic fiber is being placed on the robotic actuator, which can be moved in 4 dimensions: three spatial dimensions and angle of insertion. Of particular importance is the full MRI compatibility, which enables continuous fMRI readout during optogenetic manipulation / calcium recordings. The authors provide high-quality data on the feasibility of their approach in terms of controlling the positioning of the fiber for targeting deep brain structures. I fully agree with the assertion of the authors, that their approach represents a very important step forward, we need to tailor the position of optogenetic actuation to the particular, individual functional cytoarchitecture, particularly when targeting deeper brain regions, or for a "hypothesis-free" brain activity mapping. This work represents a mayor advance, I particularly liked the well-controlled and slow advancement of the fiber, which can drastically reduce the probability of intracerebral bleedings when passing the ventricles. The manuscript is overall well written, the figures nicely illustrate the data. However, I have one significant concern, which in my view needs to be conclusively addressed by the authors:

Mayor concern:

1. The authors make a strong and valid claim with regard to the importance of the activity-guided placement of the fiber for optogenetic actuation /calcium recordings. Yet, circuit / region specific optogenetic actuation and / or optical recordings critically also require the similarly precise injection of viral vectors encoding these actuators / indicators. The authors do not at all even mention this critical roadblock. In my view, this MgRA methodology will not lead to a mayor advance, unless paired with a similar approach for activity-guided injection. If the injection of the virus will not be guided by a similar approach, how can the authors state, that the success rate ad precision of optogenetics / opto fMRI will be drastically improved? This is particular the case for the cell and region-specific expression of optogenetic actuators in deep regions (as stated for the placement of the fiber (lines 152, but the same holds true for the injection!): the availability of strong cell specific promoters is limited, so we still need a very precise injection scheme. The authors need to address this mayor shortcoming, this paper would be much stronger, if they included an injection capability as well.

We thank the reviewer for raising this critical issue to elaborate/amplify the power of MgRA for multi-modal fMRI imaging in animals. In the revised version of the manuscript, we provide a new experiment: MgRA-driven microinjection. $MnCl_2$ solution was used as an MR contrast agent and a modified MPRAGE sequence (MdefT) [1] was used to detect the manganese-enhanced T1-weighted MRI signal (i.e. to track the injection). As previously reported, the MdefT sequence applies a 180° RF pulse with a

specific time-of-inversion (TI) to dampen the gray matter signal [2-5]. Thus, the Mdeft-based T1-weighted image can significantly increase the contrast-to-noise ratio for Mn-enhanced MRI signal with an efficient acquisition time.

As shown in **Fig. 5a**, we first targeted the central lateral thalamic nucleus (CL) and injected a Mn solution in two consecutive steps:

1). Before reaching the CL, an intermediate stop was included at the corpus callosum to deliver a small dosage of Mn^{2+} as a marker to demonstrate that we are performing a real-time microinjection using our MgRA system.

2). While the fiber tip was positioning at the CL (position verified with a T2-weighted MR image (RARE)), we performed the Mdeft sequence (~4 min) to acquire the T1-weighted images before and after the Mn injection using the same field-of-view that was used for positioning (RARE). $MnCl_2$ solution was delivered 3 times to provide dosage-dependent signal changes, demonstrating the real-time MgRA-guided microinjection procedure.

3). We also used the MgRA to drive the microinjection to the lateral hypothalamic (LH) nucleus from the same rat, further demonstrating the multi-task (multi-trial injection) potential of the MgRA-driven microinjection. As shown in **Fig. 5c-f**, the fiber tip was guided to target the LH and Mdeft images were acquired before and after the injection (3 times), showing clear effective Mn delivery to the LH. In addition, to demonstrate the reliability and spatial specificity (i.e. focal administration) of the microinjection procedure, we continuously acquired the Mdeft images ~1 hour following the injection (10 images acquired). **Fig. 5f** shows that the Mn signal is highly confined close to the injection area with limited diffusion.

In addition, this procedure has been reproduced in multiple animals and shown highly robust performance to target deep brain nuclei for the injection purpose, as quantified in **Fig 5g**. The high spatial specificity of MgRA-driven microinjection can be used to trace the brain connectivity with MEMRI [4, 5], as well as to optimize the real-time in vivo neuromodulation or molecular MRI by direct intracranial injection of drugs [6-8] and MRI contrast sensors for neurotransmitters [9-12].

It is also noteworthy that the precise fiber tip targeting to the specific functional nuclei can partially conquer the less specific viral expression of ChR2 at specific regions. In **Fig. 4**, we have shown the fiber tip-location specific functional pattern through a broad ChR2 expression in the thalamus. This scenario is comparable to optogenetic studies on transgenic mice where the ChR2 (driven by specific promoters) is expressed in multiple functional nuclei, e.g. the Tg-ChR2 mice [13-15]. Given the targeting reliability of the MgRA, it could be applied to both, not only stimulation of nuclei in animals subjected to precisely MgRA-mediated injections (animals subjected to viral injection + optogenetic stimulation by the MgRA would have reassured reliability), but also to commercially available transgenic animals, where the specificity would be given simply by the high spatial precision of the MgRA. This newly performed experiments on MgRA-driven microinjection represent a clear advantage over the standard stereotaxic-based injection procedure.

Fig.5. MgRA-driven Mn-injection into CL and LH. **a**, Top: the representative RARE anatomical image used to clarify the optical fiber location driven by MgRA for Mn injection in CC. Middle: the atlas overlapped RARE images to illustrate the fiber tip location at the CL. Eye cream is covering the craniotomy (magenta arrow). Bottom: enlarged image of fiber location. **b**, T1-weighted MPRAGE image (Mdef) showing enhanced signal from Mn injection site in the CC and CL with dose-dependency. **c**, Top: The atlas overlapped RARE images to illustrate the fiber tip location at the LH, Bottom: enlarged image of fiber location. **d**, T1-weighted MPRAGE image (Mdef) showing enhanced signal from Mn injection site in the LH with dose-dependency. **e**, Sagittal view of RARE anatomical image and MPRAGE image after MnCl₂ solution injection. **f**, T1-weighted MPRAGE image at 1 h and 1h 40 mins after the injection. **g**, the analysis of MEMRI signal at no injection, 1 injection, 2 injection and 3 injection times, as shown in b, d (n=5 injection points from 3 animals). Error bars represent mean±SD.

Minor comments:

1. Abstract: “intracellular Ca recordings...” this is misleading, as the readers will mistake this term with truly intracellular recordings such as whole-cell patch clamp. The authors conducted somatic calcium recordings

We agree with the reviewer and changed the term accordingly in the manuscript (title and main text).

2. Introduction, line 44/45: This general statement, that optic-fiber-based recordings are comparable to electrophysiology recordings is just not true. There are important differences: optical recordings of somatic calcium reflect rather suprathreshold spiking activity, while LFP recordings, which are typically combined with fMRI are dominated by subthreshold, are dominated by synaptic activity. The authors need to be more precise.

We agree with the reviewer and revised this statement in the introduction (Page 3).

3. While I fully support the notion, that the four-dimensional manipulation is a great feature, the authors need to acknowledge / discuss the limitation that it is practically impossible to change the angle / the

direction of the fiber once inserted into brain parenchyma, as this will lead to excessive tissue damage and / or bleeding. This needs to be critically discussed.

We agree with the reviewer about the limitation of MgRA. We have included a limitation section in the discussion (page 17) to highlight this issue and potential solutions to optimize the MgRA to achieve a fully automatic performance for the future studies.

Responses to reviewer #2

We thank for the encouraging and valuable comments from Reviewer #2. The reviewer stated that “Therefore, the topic of this paper is of high general interest” and “this paper provides important data, how to improve procedures and data quality in optogenetic experiments, and may thus serve to define new standards.” Here, we have revised our manuscript based on the comments and added new results to answer all the concerns from the reviewer.

This paper describes design, construction and evaluation of a MR-compatible robotic arm to position optic fibers in the rat brain for optogenetic manipulation during fMRI measurements. With the described setup fiber implantations can be monitored by real-time MRI. The authors argue that their apparatus allows for more precise positioning of the fiber than conventional stereotactic implantations, that fiber position can be corrected or altered during the experiment, and that injuries, potentially causing artefacts, can be observed before starting the actual measurement.

Optogenetic methods have recently found widespread application in many fields of neuroscience. In particular the combination with fMRI and optical readout methods has shown great potential for extending our understanding of brain wide and specific neural responses. Therefore, the topic of this paper is of high general interest. Although the direct reproducibility of this work (ie copying the robotic arm) appears not easily feasible for other labs, this paper provides important data, how to improve procedures and data quality in optogenetic experiments, and may thus serve to define new standards. The authors provide data of outstanding quality in terms of (implantation) precision, image resolution and sensitivity of both BOLD and calcium signal. The data fully support the claim that the proposed procedure is superior to conventional stereotactic implantation. Also the gain in data quality and signal level by placing the fiber under MR guidance is impressively illustrated. Further, this paper provides data on neuronal and hemodynamic network response to optogenetic stimulation, which is still not too abundant in the literature.

The paper is well structured and written – it was a pleasure to read.

A few minor issues remain, before this paper may be accepted for publication:

1. Not all components of the robotic arm are described in detail (eg the stepper motor, camera,...). It might be helpful to add a list of all components in the Supplement.

We have revised our description of MgRA in **METHODS** (Page 19) and **Supplementary Fig. S12, S13**, as well as added a list of all components in the Supplementary information.

Figure S12. Detailed design of the MgRA (figures from the approved European patent). a, The schematic view of the whole MgRA mechanical design including the cross table to mount the stepper motors. b, The coupling of the stepper motors (back part) to the matching toothed pulley in the head was achieved by a synchronous belt drive in a form-fit manner. c, Custom-designed rat holder with a built-in MRI compatible camera, surface coil and head part of the MgRA. d, The components of the head part of the MgRA. For more details see the approved European patent as following link: (<https://patentscope.wipo.int/search/en/detail.jsf?docId=EP215319263&tab=PCTDESCRIPTION&maxRec=1000>).

2. Naming of products and companies is incoherent. This should be in a unified format, eg (product, company, country)

We have modified the manuscript to keep names of products and companies coherent.

3. Line 345: specify: “sealed with bone”

We have specified that the craniotomies were sealed with the bone wax.

4. Line 360: which adhesive gel was used

We have modified that ‘An adhesive gel was used to secure the calcium recording fiber to the skull’.

5. Line 397-402: Only anatomical MRI with resolution of 150 μm is described. What about the high resolution images in Fig. 1?

Higher resolution (50 μm) RARE sequence, specifically for **Fig. 1**, to accurately identify the optical fiber in the coronal plane, with the following parameters: Repetition time, 1500 ms; Echo Time, 11.0428 ms; FOV: 1.92cm × 1.56 cm, matrix = 384 × 312, resolution = 50 μm × 50 μm, slice thickness = 0.75 mm,

RARE factor = 6, averages = 6. We have added the description for the parameters in **Fig.1** to **METHODS** (Page 22).

6. Line 427 and following unnumbered line: In both equations t has been replaced by 4. Please correct.

We thank for the important notes from the reviewer. We have corrected it.

I noticed two extra points, which do not appear relevant for the message of the paper. Yet, I am curious to have the authors' comments:

7. In Fig 4, Fig S7, and Fig S8 calcium signal is almost 100 %. Why is this so much higher than in the other measurements?

We thank for the reviewer for raising this question. The percentage change of GCaMP-mediated calcium signal is mainly based on the baseline of the fiber optic recording, which can be contributed by many factors, including the GCaMP expression level, the micro-environment of the cortex surrounding the fiber tip, as well as the efficiency of the optical fibers to transmit the optical signal from the brain to the photomultiplier. One plausible explanation of this salient difference of our measurements from different animals may mainly come from the varied micro-environment of the cortex, e.g., the micro-bleeding caused by the fiber insertion may potentially lead to altered responses in the cortex upon optogenetic stimulation. In this manuscript, we are mainly focused on evaluating the region-specific activation patterns from the MgRA-guided optogenetics. Although the measurements across animals showed large variability, the corresponding differences to the MgRA-driven region-specific activation shows highly reliable readouts, which can still be used to verify our method development.

8. In Fig S8b (3 Hz) and Fig S10d (lower 3 positions) every other calcium response is suppressed/reduced. Similar phenomena are also present in other publications. Yet, I am not aware of an explanation for this variation. Can the authors comment on that.

We thank the reviewer for highlighting this interesting observation. In our previous studies, we have shown that the evoked neuronal calcium spikes are highly correlated to the LFP spikes with the electrical stimulation of the forepaw or whisker pad[16]. This interleaved amplitudes of calcium spikes were also observed in LFP recordings. Thus, it is not a calcium-specific effect. We think that this is more close to the desensitization effects following the long-train of electrical stimulation pulses. As shown in **Supplementary Fig. S 8**, there is highly robust spikes detectable at the lower frequency since there is sufficient time for the neural circuits to recover from the previous pulse, but the responses can be possibly suppressed when pulses are so close to each other. Thus, for the 3Hz, it may show interleaved amplitude patterns, i.e. the recovery occurs longer than 300ms before the third pulse comes, but at the even higher frequency, we could observe a more suppressed responses since the high frequency pulses deplete the glutamate constantly and do not provide sufficient time for the recovery.

In summary, I consider this as an excellent piece of work, which should be published after these minor issues have been addressed.

Responses to reviewer #3

We thank for the valuable comments from reviewer #3. We have revised our manuscript based on the comments and suggestions from the reviewer. In particular, we apologize not to clarify the special

needs and critical context for the development of the proposed MgRA system for high field MRI scanner with limited running space, which is crucial for animal brain functional mapping in combination with optogenetics and optical fiber calcium imaging, i.e. the multi-modal fMRI platform.

This review is specifically focused on the innovation of the MRI- guided robotic arm (MrGA).

While the authors clearly have a unique application and there is certainly great benefit to an MRI-compatible robotic arm, the device presented does not have the novelty the authors indicate and the review of related literature is completely missing. If the focus of the paper was on what is done with the laser delivery, then this may not be a problem. But, the authors propose the MrGA as a key novel aspect and this is just not true.

Specifically, in line 69 of p.4 the authors state: "there is currently no MRI-compatible, automatic robotic control system to assist fiber optic insertion for multi-modal fMRI platforms." This is simply not true, in fact there is a whole host of research and devices (some in clinical and preclinical trials) for MRI-compatible robotic devices for stereotactic neurosurgery.

We have revised the statement and highlight the novelty of our work in the context of the multi-modal fMRI using the high field MRI scanner with limited running space (Introduction Page 4).

Two examples of research systems in preclinical trials are:

<https://www.worldscientific.com/doi/abs/10.1142/S2424905X18500034>

<https://academic.oup.com/neurosurgery/advance-article-abstract/doi/10.1093/neuros/nyy266/5037802>

There are also clinical systems for stereotactic neurosurgery (manual and robotic):

<https://www.mriinterventions.com/clearpoint/clearpoint-overview>

<https://www.monteris.com/neuroblate-system/>

In the rehabilitation robotics community, robots have been used in MRI w/ fMRI such as:

<https://ieeexplore.ieee.org/document/1618680>

Some of these systems are for delivering ultrasonic or fiberoptic probes, typically for thermal ablation or temperature monitoring. They can be used with multi-modal MRI including typical anatomical imaging sequences, thermal imaging, functional imaging, etc.

While these may not be specific to the author's specific application, the authors should compare the novelty of their approach and frame their contributions appropriately (which appears primarily to fit inside a small bore ultra high field scanner).

We thank for the critical and valuable comments from the reviewer. We have revised our introduction and discussion to highlight the existing state-of-the-art MRI-compatible robotic devices for stereotactic neurosurgery.

With regard to the robotic device, the description could use more detail. In particular, it is not clear how the motion is measured. Stepper motors are inherently likely to slip and miss steps, especially if unpowered (which would presumably be the case during as much of the imaging time as possible to minimize electrical noise and vibration). Have the investigators looked into using encoders or another measurement approaches?

We agree with the reviewer that the performance of stepper motors is critical to achieve high precision and stability of the robotic arm. Therefore, an encoder (NOE2-05-B14, Nanotec, Germany) is used with

motor controller (SMCI33-1, Nanotec, Germany) so that the stepper motor (ST4118D1804-B, Nanotec, Germany) can be run in a close-loop mode, as shown in **Supplementary Fig. S13 a,b**. The resolution of the encoder is 1000 CPR.

Also noteworthy is that our MgRA system allows the stepper motors to work in full powered mode during MRI scanning since they are positioned outside of the scanner. The unpowered condition is usually avoided during the targeting procedure. After the optical fiber was precisely positioned to the region of interests, the motor will be unpowered. For the sudden unpowered case, there is an integrated function in the custom-designed software which is to send a ready-inquiry command before any programmed procedures can be resumed. If the motor or the controller is unpowered, then there will be an error message to warn the operator.

Figure S13. Detailed design of the back part of the MgRA. a The coupling of the stepper motors (back part) to the matching toothed pulley in the head was achieved by a synchronous belt drive in a form-fit manner. b The encoder (NOE2-05-B14, Nanotec, Germany) is used with motor controller (SMCI33-1, Nanotec, Germany) so that the stepper motor (ST4118D1804-B, Nanotec, Germany) can be run in a close-loop mode. c Multi-groove belt (optibelt OMEGA 3M, Optibelt, Germany) used to fit into a matching toothed pulley. d Closed belts can be cascaded to transfer the motion (red arrow). All schematic figures shown here are from the approved MgRA European patent.

Are there issues with belt stretch, backlash, slippages that affect the very high stated accuracy (not accuracy study was shown). Was there an assessment of the affect of the motors (electrical noise, vibration, etc) on image quality?

We thank for the thoughtful comments from the reviewer. In the Discussion section, we have highlighted the potential issues and provide possible strategies to solve the related issues. Here, we highlighted several key efforts to reduce the raised concerns:

Firstly, we used high quality multi-groove belts (optibelt OMEGA 3M, Optibelt, Germany) with specially designed teeth which fit into a matching toothed pulley (**Supplementary Fig. S13 c**). When correctly tensioned, they have no slippage, and run at constant speed.

Secondly, instead of one single long close belt from the stepper motor to the pulley in the head part of the robotic arm, up to four closed belts can be cascaded to transfer the motion (**Supplementary Fig. S13 d**). Therefore, the driving mechanism can be easily adjusted to the tension and the space available in the scanner room by altering the belt length.

Thirdly, gearbox (GPLE22-2S-12, Nanotec, Germany) is used to decrease the speed, reduce the ratio (Reduction Ratio: 12) and increase the output torque nominal (1.5 Nm) to make the movement smoothly (**Supplementary Fig. S13 b**). Also, the MRI compatible cameras are used in the scanner to monitor the movement of the fiber (**Fig. 2 a,b and Supplementary Movie S5**).

Since the MgRA system positioned the motors outside of the MRI scanner, not only the motor can fully function without the magnetic interference, but also the MRI imaging is not distracted by the motor performance. In **Supplementary Movie S2, S3**, we can detect highly reliable movement of the optical fiber tip without potential motor interference.

There also does not seem to be any discussion of registration to know where the coordinates of the robot arm are relative to the imaging system (and the subject).

We have revised the method and discussion section to provide more detailed information on the registration issue.

Agarose has been previously applied above the burr hole of the skull and the fiber tip (previously positioned above the burr hole using the MgRA system under the guidance of the build-in camera inside the MR scanner) can be directly imaged to determine its coordinates in the MRI images (**Supplementary Fig. S3**). Then, we developed the algorithm to register four coordinate system for the fiber tip position: Brain atlas coordinates (Co1, Paxinos & Watson rat brain atlas, 6th edition), MRI/DTI rat brain atlas (Co2, provided by Dr. G. Allen Johnson), MRI coordinate (Co3) and robotic arm coordinate (Co4). In short, the Co1 is first transferred to the Co2 by the algorithm (**Supplementary Fig. S3 b, c**). By registering the 2D anatomical images of individual rat (Co3) to the MRI/DTI brain atlas (Co2), the transformation between the Co1 and Co3 is settled (Red arrow in **Supplementary Fig. S3 b, c, d**). Since the fiber tip position is directly detected in the MRI images above the craniotomy, the related coordinate offset from the fiber tip to the targeted function nuclei can be calculated based on the multiple transformation matrices.

In the discussion section, we also described the potential limitation of this method. Although we acquired the 3D anatomical images of the rat brain, the major registration procedure between atlas and MRI images is still based on a 2D registration algorithm, which is applied to control the fiber tip movement along the dorsal-ventral direction. In the future development, we will try to provide a 3D registration system so that we could take advantage of the full motor control movement capability of the MgRA system.

Figure S3. MRI-based relocation outside of the rat brain and the registration of coordinates. a Agarose with manganese was applied to cover the skull (yellow arrow). By lowering the fiber into the agarose, we could calculate the distance between fiber tip (green arrow) and burr hole (red arrow) from the anatomical images. The burr hole is filled with agarose as well. b Brain atlas coordinates (Co1)[17]. c MRI/DTI Atlas of the Rat Brain (Co2, provided by Dr. G. Allen Johnson)[18]. d 3D anatomical images of an individual rat.

In summary, I think the authors need to do a much better job of framing the novel contributions of the work. What is presented is strangely in the discussion at the end of the paper and not up front in the introductions and/or the description of the device. What they are doing with the probe is very interesting, but the robotic devices does not represent any real novel contribution to science and is in fact much less capable than many existing research platforms in the neurosurgery space (which the authors make no attempt to review). Just to clarify, please note that is not to say it is not a well-designed device or that is is not of great utility.

We have revised the introduction based on the suggestions from the reviewer.

REFERENCES

1. Mugler, J.P., 3rd and J.R. Brookeman, *Three-dimensional magnetization-prepared rapid gradient-echo imaging (3D MP RAGE)*. Magn Reson Med, 1990. **15**(1): p. 152-7.
2. Yu, X., et al., *Thalamocortical inputs show post-critical-period plasticity*. Neuron, 2012. **74**(4): p. 731-42.
3. Yu, X., et al., *Deciphering laminar-specific neural inputs with line-scanning fMRI*. Nat Methods, 2014. **11**(1): p. 55-8.

4. Pautler, R.G. and A.P. Koretsky, *Tracing odor-induced activation in the olfactory bulbs of mice using manganese-enhanced magnetic resonance imaging*. Neuroimage, 2002. **16**(2): p. 441-8.
5. Pautler, R.G., A.C. Silva, and A.P. Koretsky, *In vivo neuronal tract tracing using manganese-enhanced magnetic resonance imaging*. Magn Reson Med, 1998. **40**(5): p. 740-8.
6. Turchi, J., et al., *The Basal Forebrain Regulates Global Resting-State fMRI Fluctuations*. Neuron, 2018. **97**(4): p. 940-952 e4.
7. Saleem, K.S., et al., *Magnetic resonance imaging of neuronal connections in the macaque monkey*. Neuron, 2002. **34**(5): p. 685-700.
8. Jaime, S., et al., *Delta Rhythm Orchestrates the Neural Activity Underlying the Resting State BOLD Signal via Phase-amplitude Coupling*. Cereb Cortex, 2019. **29**(1): p. 119-133.
9. Okada, S., et al., *Calcium-dependent molecular fMRI using a magnetic nanosensor*. Nat Nanotechnol, 2018. **13**(6): p. 473-477.
10. Ghosh, S., et al., *Probing the brain with molecular fMRI*. Curr Opin Neurobiol, 2018. **50**: p. 201-210.
11. Hai, A., et al., *Molecular fMRI of Serotonin Transport*. Neuron, 2016. **92**(4): p. 754-765.
12. Lee, T., et al., *Molecular-level functional magnetic resonance imaging of dopaminergic signaling*. Science, 2014. **344**(6183): p. 533-5.
13. Carter, M.E., et al., *Tuning arousal with optogenetic modulation of locus coeruleus neurons*. Nat Neurosci, 2010. **13**(12): p. 1526-33.
14. Kunwar, P.S., et al., *Ventromedial hypothalamic neurons control a defensive emotion state*. Elife, 2015. **4**.
15. Meek, T.H., et al., *Functional identification of a neurocircuit regulating blood glucose*. Proc Natl Acad Sci U S A, 2016. **113**(14): p. E2073-82.
16. Wang, M., et al., *Brain-state dependent astrocytic Ca(2+) signals are coupled to both positive and negative BOLD-fMRI signals*. Proc Natl Acad Sci U S A, 2018. **115**(7): p. E1647-E1656.
17. Paxinos, G. and C. Watson, *The rat brain in stereotaxic coordinates*. 6th ed. 2007, Amsterdam ; Boston ;: Academic Press/Elsevier.
18. Paxinos, G., et al., *MRI / DTI atlas of the rat Brain*. 2015, London ; San Diego: Elsevier, Academic Press. xxix pages.

Reviewers' comments:

Reviewer #1 (Remarks to the Author):

The authors comprehensively addressed my concerns, they included new data showing MgRA-driven microinjections. In my view, this significantly strengthened the manuscript. I fully support publication.

Reviewer #2 (Remarks to the Author):

The authors have addressed all points I had raised and have further improved their manuscript. Adding a section that shows the capability of performing injections with the robotic arm, has further strengthened the paper.

Elaborating more on the differences compared to existing robotic microsurgical devices, has made the technical challenges and achievements of this work even more tangible. Although existing robotic devices may provide higher precision and accuracy, none of those was able to operate in such limited space at high magnetic field. For experimental multimodal neuroimaging this robotic arm will enable new experimental designs and dramatically improve quality and reproducibility of stereotaxic procedures during MRI.

Cornelius Faber, University of Muenster, Germany

Reviewer #3 (Remarks to the Author):

Overall the authors seem to have addressed the reviewer feedback.

In particular, with regard to the MRI-compatible robot arm the authors significantly improved the description with respect to related prior art and clarified the contributions.

The authors also include more detail about the robotic system design and the registration aspects in the Supplemental material.

Of minor note, the authors use the term "MRI compatible" for their robot, but it does not seem to be validated. Typically MRI compatible robotic systems would report information such as changes in image quality (including SNR, warping, etc) with and without the robot. However, if the authors can demonstrate adequate image quality for their need, perhaps that is sufficient for this case.

Responses to reviewers

Reviewer #1:

The authors comprehensively addressed my concerns, they included new data showing MgRA-driven microinjections. In my view, this significantly strengthened the manuscript. I fully support publication.

Reviewer #2

The authors have addressed all points I had raised and have further improved their manuscript. Adding a section that shows the capability of performing injections with the robotic arm, has further strengthened the paper.

Elaborating more on the differences compared to existing robotic microsurgical devices, has made the technical challenges and achievements of this work even more tangible. Although existing robotic devices may provide higher precision and accuracy, none of those was able to operate in such limited space at high magnetic field.

For experimental multimodal neuroimaging this robotic arm will enable new experimental designs and dramatically improve quality and reproducibility of stereotaxic procedures during MRI.

Reviewer #3

Overall the authors seem to have addressed the reviewer feedback.

In particular, with regard to the MRI-compatible robot arm the authors significantly improved the description with respect to related prior art and clarified the contributions.

The authors also include more detail about the robotic system design and the registration aspects in the Supplemental material.

Of minor note, the authors use the term "MRI compatible" for their robot, but it does not seem to be validated. Typically MRI compatible robotic systems would report information such as changes in image quality (including SNR, warping, etc) with and without the robot. However, if the authors can demonstrate adequate image quality for their need, perhaps that is sufficient for this case.

We thank for the thoughtful comment from the reviewer. For the developed MgRA system, all the components for the head-parts were constructed from fully MRI-compatible materials and the MgRA-based positioning can be performed in real time during scanning with a long mechanical arm. In our future work, we will develop the miniaturized piezo-motor based robotic system, which will make the comparison of image quality due to the motor-related interference more valuable inside the high field MRI scanner, as previously reported in a lower magnetic field strength^{1,2}.

REFERENCES

1. Shokrollahi, P., Drake, J.M. & Goldenberg, A.A. Ultrasonic motor-induced geometric distortions in magnetic resonance images. *Med Biol Eng Comput* **56**, 61-70 (2018).
2. Shokrollahi, P., Drake, J.M. & Goldenberg, A.A. Signal-to-noise ratio evaluation of magnetic resonance images in the presence of an ultrasonic motor. *Biomed Eng Online* **16**, 45 (2017).